# Guidelines' recommendations for the treatment-resistant depression: A systematic review of their quality

**Franciele Cordeiro Gabriel**[1][©]*, **Airton Tetelbom Stein**[2,3][©], **Daniela de Oliveira Melo**[4][©], **Gessica Caroline Henrique Fontes-Mota**[1][‡], **Itamires Benício dos Santos**[4][‡], **Camila da Silva Rodrigues**[4][‡], **Andrea Dourado**[4][‡], **Mônica Cristiane Rodrigues**[5][‡], **Renério Fráguas**[6][‡], **Ivan D. Florez**[7,8,9][©], **Diogo Telles Correia**[10][‡], **Eliane Ribeiro**[1][©]

**1** Departamento de Farmácia, Faculdade de Ciências Farmacêuticas, Universidade de São Paulo, São Paulo, São Paulo, Brasil, **2** Departamento de Saúde Coletiva, Universidade Federal de Ciências da Saúde de Porto Alegre, Porto Alegre, Rio Grande do Sul, Brasil, **3** Gerência de Ensino e Pesquisa do Grupo Hospitalar Conceição, Porto Alegre, Rio Grande do Sul, Brasil, **4** Departamento de Ciências Farmacêuticas, Instituto de Ciências Ambientais, Químicas e Farmacêuticas, Universidade Federal de São Paulo, Diadema, São Paulo, Brasil, **5** Faculdade de Saúde Pública, Universidade de São Paulo, São Paulo, São Paulo, Brasil, **6** Departamento e Instituto de Psiquiatria, Hospital das Clínicas, Faculdade de Medicina da Universidade de São Paulo, Divisão de Psiquiatria e Psicologia Hospital, São Paulo, Brasil, **7** School of Rehabilitation Sciences, McMaster University, Hamilton, Ontario, Canada, **8** Department of Pediatrics, University of Antioquia, Medellín, Colombia, **9** Pediatric Intensive Care Unit, Clinica Las Americas-AUNA, Medellin, Colombia, **10** Departamento de Psiquiatria e Psicologia da Faculdade de Medicina da Universidade de Lisboa, Lisboa, Portugal

© These authors contributed equally to this work.
‡ GCHFM, IBS, CSR, AD, MCR, RF and DTC also contributed equally to this work.
* francordegabriel@gmail.com

**Data Availability Statement:** All relevant data are within the paper and its Supporting information files.

## Abstract

### Introduction

Depression is a serious and widespread mental health disorder. A significant proportion of patients with depression fail to remit after two antidepressant treatment trials, a condition named treatment-resistant depression (TRD). Clinical practice guidelines (CPGs) are instruments aimed to improve diagnosis and treatment. This study objective is to systematically appraise the quality and elaborate a comparison of high-quality CPGs with high-quality recommendations aimed at TRD.

### Methods and analysis

We searched several specialized databases and organizations that develop CPGs. Independent researchers assessed the quality of the CPGs and their recommendations using AGREE II and AGREE-REX instruments, respectively. We selected only high-quality CPGs that included definition and recommendations for TRD. We investigated their divergencies and convergencies as well as weak and strong points.

### Results

Among seven high-quality CPGs with high-quality recommendations only two (Germany's Nationale Versorgungs Leitlinie–NVL and US Department of Veterans Affairs and

**Funding:** F.C.G. is a fellow of the Conselho Nacional de Pesquisa e Tecnologia (CNPq, grant number 141811/2020-0).

**Competing interests:** I have read the journal's policy and the authors of this manuscript have the following competing interests: Ivan Florez is the current leader of the AGREE collaboration.

Department of Defense–VA/DoD) included specific TRD definition and were selected. We found no convergent therapeutic strategy among these two CPGs. Electroconvulsive therapy is recommended by the NVL but not by the VA/DoD, while repetitive transcranial magnetic stimulation is recommended by the VA/DoD but not by the NVL. While the NVL recommends the use of lithium, and a non-routine use of thyroid or other hormones, psychostimulants, and dopaminergic agents the VA/DoD does not even include these drugs among augmentation strategies. Instead, the VA/DoD recommends ketamine or esketamine as augmentation strategies, while the NVL does not mention these drugs. Other differences between these CPGs include antidepressant combination, psychotherapy as a therapeutic augmentation, and evaluation of the need for hospitalization all of which are only recommended by the NVL.

## Conclusions

High-quality CPGs for the treatment of depression diverge regarding the definition and use of the term TRD. There is also no convergent approach to TRD from currently high-quality CPGs.

## Introduction

Depression is a serious medical illness that negatively affects behavior. It is a common condition, affecting more than 300 million people worldwide and it is considered one of the most relevant public health problems in the 21st century [1]. Owing to its disabling nature, it can cause various professional, economic, social, and personal losses [2]. In addition, the number of people with depression has increased considerably over the last few years, overloading health systems and generating a greater need for resource optimization [3]. Patients with lower response to depression treatment have a higher risk of severe outcomes including job loss, isolation, and suicide, which may lead to an increased economic cost to society [4].

There are several effective classes of antidepressants including selective serotonin reuptake inhibitors (SSRIs), serotonin and norepinephrine reuptake inhibitors (SNRIs), tricyclic antidepressants, noradrenergic and serotonergic antidepressants, monoamine oxidase inhibitors, among others [5]. Around only one-third of patients will remit after treatment with an SSRI and only 25% to 27% will remit after a subsequent treatment trial with another antidepressant [6]. Consequently, a significant number of patients—up to 40%—will be classified as having treatment-resistant depression (TRD; i.e., they failed to remit after at least two antidepressant treatment trials) [6].

TRD is difficult to manage, and results are usually poor, especially when unstandardized approaches are used [7–10]. In this regard, clinical practice guidelines (CPGs) are essential tools for guiding clinical decisions. CPGs facilitate treatment standardization and are supported by the best available evidence gathered through rigorous systematic reviews. Moreover, high-quality CPGs also consider elements that are key before recommending interventions, such as costs, acceptability, feasibility, patients' values and preferences, and the balance between benefits and harms [11, 12].

Unfortunately, in a recent review, only 4 out of 11 available CPGs for the treatment of depression met the minimum quality benchmark after their assessment with the AGREE II

tool [13]. Another review found that only 6 out of 27 CPGs could be classified as high-quality guidelines [14].

Low-quality guidelines increase the likelihood of bias in the development of recommendations, which in turn can cause inconsistencies among recommendations [15, 16]. In a study comparing the recommendations for pharmacotherapy and neurostimulation in the treatment of depression, a high degree of inconsistency was found in the recommendations for the second and third lines of treatment [17]. Discrepancies in recommendations for first-line treatments and patients not responding to first-line treatment were also observed by MacQueen et al. [16], in a study analyzing CPGs used in primary healthcare for major depression, dysthymia, and minor depression.

High-quality CPGs contain recommendations are based on the best evidence, with a transparent and trustworthy process that is implementable and acceptable by stakeholders and patients. However, it should be noted that even high-quality CPGs, i.e., CPGs that obtain high scores on AGREE II assessment, do not necessarily guarantee credible and implementable recommendations [18]. In response to this gap between the quality of the guideline and the quality of recommendations, the AGREE Collaboration developed another instrument to evaluate the quality of the recommendations: the Appraisal of Guidelines Research and Evaluation-Recommendations Excellence—the AGREE-REX [19, 20]. AGREE-REX includes additional factors, such as clinical applicability, values and preferences, and implementability. These factors are considered by guideline developers and users to be associated with more feasible, credible, and implementable recommendations.

Previous studies have evaluated the methodological quality of CPGs for the treatment of depression using AGREE II and the discrepancies among recommendations [14, 21–23]. However, currently, no study has specifically evaluated the quality of CPGs' recommendations using the AGREE-REX instrument. To enhance evidence-based clinical practice implementation, it is essential to evaluate quality to help increase the reliability and applicability of recommendations. There is a particular need to have trustworthy guidelines for TRD. Thus, this systematic review aimed to identify and appraise the recommendations for the approach of TRD in CPGs.

## Materials and methods

This study is a systematic review conducted across the following databases: MEDLINE (through PubMed), Embase, Cochrane CENTRAL, PsycINFO, and the Virtual Health Library. The complete database search strategy is presented in the S1 File. A manual search was also undertaken on the websites of organizations responsible for developing or compiling CPGs. The protocol for this systematic review has already been published [24]. The search was made in December 2021 and covers the period of January 2011 to December 2021. Then, in June 2022, we searched the literature to update the included CPGs.

Briefly, we included all documents that provided recommendations for the pharmacological management of treatment-resistant depression, regardless of the methodology used. We excluded CPGs that did not specifically refer to TRD, were intended for local use or were related to patients with specific comorbidities.

After excluding duplicates, titles and abstracts retrieved from the searches were independently screened by two reviewers to verify their eligibility. The documents were checked for their eligibility in full text (two full texts could not be retrieved).

Eligible references were obtained in full text and reviewed in duplicate and independently by two reviewers. Data extraction was also performed independently and in duplicate by two reviewers. Characteristics extracted from each CPG included: the year of publication or an

updated version of the CPGs, classification of evidence, and institution or organization. This task also involved extracting the quality of the evidence and strength of the recommendations. Disagreements were resolved by consensus, or by a third researcher in all the previous steps.

The quality of the CPGs and their recommendations were evaluated using AGREE II [25, 26] and AGREE-REX, respectively [18]. Details about these instruments and how they were applied have been explained elsewhere [24].

Only CPGs considered to have high-quality with recommendations also categorized as high-quality were included in the data synthesis. CPGs with a score higher than 60% in domain 3 (rigor of development) of AGREE II were classified as high-quality. Recommendations with a score higher than 60% in domain 1 (clinical applicability) of AGREE-REX were classified as high-quality. Recommendations were grouped into two main topics, "terminology for TRD" and "recommended management strategies for TRD". The terminologies and sequences of the therapeutic strategies were compared between the CPGs and their agreement and disagreement were synthesized in a table (Table 1).

We adopted the definition of TRD as failure to respond after at least two antidepressant treatment trials with adequate dose and time of treatment as has been used in the STAR*D Trial [6].

**Table 1. Recommendations for the management of TRD obtained from the selected CPGs.**

| Recommendations | NVL CPG | VA/DoD CPG |
|---|---|---|
| | 2015 [33] | 2022 [32] |
| Augmentation drugs | R | R |
| Sequenced augmentation | NM | NM |
| Lithium | R# | NM |
| Thyroid hormone or another hormone | R* | NM |
| Psychostimulants | R* | NM |
| Dopamine | R* | NM |
| Ketamine | NM | R |
| Esketamine | NM | R |
| Switch medication | NM | NM |
| Sequence among switch | NM | NM |
| Mention specific drugs/classes | NM | NM |
| Combination | R# | NM |
| Sequence among combination | NM | NM |
| Mention specific drugs/classes | NM | NM |
| Neuromodulation | R | R |
| Sequence among neuromodulation | NM | NM |
| Electroconvulsive therapy | R | NR |
| Repetitive transcranial magnetic stimulation | NM | R |
| Deep brain stimulation | NM | NR |
| Vagal nerve stimulation | NM | CI |
| Psychotherapy | R | NM |
| Sequence among psychotherapy | NM | NM |
| Type | NM | NM |
| Evaluate the need for hospitalization | R | NM |

R = recommended; NR = not recommended; NM = not mentioned in the text; CI = mentioned in the text but contraindicated in the recommendations;

* should not be used routinely;

# not listed in the recommendations section, but mentioned in the CPG text.

This definition, although not universally accepted, is commonly used, and is recognized by practitioners and major organizations as a term referring to a specific clinical condition.

## Results

We retrieved 5,063 documents from the searches, removed 419 duplicates, and ended up with 4,644 references. We discarded nonrelevant references by the title and abstract and retrieved 174 works for checking their eligibility in full text (two full texts could not be retrieved). A total of 126 documents were excluded and 48 documents were included after the full-text review. We also identified 15 documents from the guidelines' repositories. Finally, 63 CPGs were selected. Of these 63 CPGs, 17 were classified as high-quality, and among them seven with high-quality recommendations. The flowchart showing this selection process is presented in S1 Fig. Here we examined these seven CPGs with high-quality recommendations. From these 7, five [27–31] were excluded since they did not specifically refer to TRD. The reasons for the exclusion of each CPG were: the CPG from the National Institute for Health and Care Excellence—NICE, UK, 2022 [28] uses a stepped-care model, not involving specifically the concept of TRD to develop their recommendations; the CPG from the Spain Working Group, 2011 [31]: uses the term "depression resistance" and their recommendations were intended for an inadequate response after failure to one therapeutic trial and not after two trials; the CPG of the Colombian, Ministerio de Salud, 2013 [29]: only brings recommendations based on treatment failure after just one treatment trial instead of considering the failure after two trials; the Peru EsSalud, 2019 CPG [27]: focuses only on mild depression and does not address TRD; and the CPG from the American Psychological Association, 2019 [30]: has recommendations to partially responsive or no-responsive patients but does not use the TRD definition.

We included only two CPGs: the VA/DoD, Department of Veterans Affairs and Department of Defense, US (2022) [32], and the National Supply Guideline (Nationale Versorgungs Leitlinie—NVL) from Germany (2015) [33], which provided recommendations specifically elaborated for TRD defined as failure to respond to at least two pharmacological treatment trials with adequate dose and time of treatment (Table 1).

Although the CPGs from VA/DoD (2022) [32] and NVL (2015) [33] are similar regarding their high-quality recommendations, there they have some important divergencies. Of relevance, regarding augmentation with neuromodulation strategies, electroconvulsive therapy (ECT) is recommended by the NVL CPG [33], but not by the VA/DoD CPG [32], while repetitive transcranial magnetic stimulation is recommended by the VA/DoD CPG [32] but is not mentioned by the NVL CPG [33]. Differences also exist regarding augmentation with drugs. While the NVL CPG [33] recommends the use of lithium, and a non-routine use of thyroid or other hormones, psychostimulants, and dopaminergic agents the VA/DoD CPG [32] does not include these drugs among the augmentation strategies for TRD. Instead, the VA/DoD CPG [32] recommends ketamine or esketamine as augmentation strategies for TRD, while NVL CPG [33] does not include these drugs. Other differences between VA/DoD [32] and NVL [33] CPGs include antidepressant combination as a strategy for TRD, recommended only by NVL CPG [33]; psychotherapy as a therapeutic augmentation option recommended only by NVL CPG [33]; and evaluation of the need for hospitalization, recommended only by the NVL CPG [33].

## Discussion

From the seven selected CPGs that contained high-quality recommendations, only two presented recommendations specifically addressing TRD based on the definition we adopted here. Therefore, we examined only these two CPGs, the VA/DoD from the Department of

Veterans Affairs and Department of Defense, US [32], and the NVL from Germany [33]. Both had recommendations specifically elaborated for TRD defined as failure to respond to at least two pharmacological treatment trials with adequate dose and time of treatment. The remaining five high-quality CPGs did not specifically refer to TRD, one focused only on mild depression [27], one used a stepped-care model and a dimensional concept of responsiveness [28], two CPGs considered treatment failure after just one trial [29, 31], and one used the denomination partially responsive or no-responsive patients but did not use the TRD definition [30]. The NICE guideline [28] was updated in June 2022 to include interventional procedures guidance for "Implanted vagus nerve stimulation for treatment-resistant depression". Even though in this specific recommendation the authors use the term "treatment-resistant", they defend that this term should not be used and in the guideline as a whole, they adopted a stepped-care model replacing the concept of resistance by the level of response to treatment.

## The concept of treatment-resistant depression

There is no consensus regarding the definition and use of the term TRD. Several authors have underlined the lack of a common or consensual terminology for treatment responsiveness of depression [34, 35]. The VA/DoD CPG [32] clearly defined treatment resistance following the STAR*D Trial, which considers TRD as a therapeutic failure after two adequate antidepressant therapies [6]. Although the CPG from NVL [33] did not clearly provide a single definition, they recognize the above definition as one of the most used [32]. Despite this definition being frequently used during the last decades, the challenge to achieve its widespread acceptance continues. It has been suggested that the reasons for that include the absence of standardized criteria to consider what constitutes an adequate treatment trial failure and if non-pharmacological treatments should be considered or not [36].

Five of the seven selected high-quality CPGs did not adopt the concept of resistant depression as described in the present article. The CPG by the Ministry of Health, Social Services and Equality from Spain (2014) [31] mentions the lack of a universally accepted definition for TRD, which makes it difficult to interpret findings and this leads to limitations in applying evidence-based recommendations. This CPG has a specific topic for resistant depression, although considers it as one that partially or does not respond to treatment, without requiring a minimum of two treatment failures [31]. The NICE 2022 CPG from the United Kingdom [28], abandoned the term "resistance" defending that it was perceived as pejorative, and not supported by evidence [28]. Instead, it recommends a stepped-care model replacing the concept of resistance with the level of response to treatment. The APA CPG [30] does not address resistant depression and uses the terms partial or non-responders.

This lack of consensus on definitions for resistant depression complicates the designing of clinical trials and may impair the care given to depression patients. Researchers, health professionals, and patients would benefit greatly from a standard definition of resistant depression.

## Differences and similarities of recommendations from included CPGs

Both available CPGs that provide high-quality recommendations for TDR differ in some respects.

The VA/DoD [32] recommends ketamine or esketamine as drug augmentation strategies for TRD based on its support by a systematic review [37]. However, they recommend caution with the use of these medications since studies with better methodology, low risk of bias, and longer follow-ups are needed. The CPG from NVL [33], recommends drug augmentation for patients who do not respond to antidepressant treatment, but without requiring the diagnosis of TRD, including lithium, quetiapine, aripiprazole, olanzapine or risperidone [27–33]. On the

other hand, this CPG [33] recommends not use routinely carbamazepine, lamotrigine, pindolol, valproate, dopamine agonists, psychostimulants, or thyroid hormones as augmentation strategies after antidepressant treatment failure to unipolar depression.

The drug combination is briefly cited as a valid option for TRD by the CPG from NVL [33]. However, this CPG does not provide recommendations about which specific antidepressants or antidepressant classes could be combined. The VA/DoD CPG [32] does not mention the combination of antidepressant drugs neither for TRD nor for patients who presented a partial or limited response to initial treatment.

Both CPGs include drug switching as a treatment option after the first antidepressant failure, which is in line with the finding that most CPGs recommend drug switching before drug augmentation or a combination [38]. They do not mention antidepressant switching among the recommendations for TRD.

Among neuromodulation and other somatic strategies, the CPG from NVL [33] recommends electroconvulsive therapy but does not mention transcranial magnetic stimulation. The VA/DoD CPG [32] mentions various somatic strategies for patients with TRD including deep brain stimulation, electroconvulsive therapy, transcranial magnetic stimulation, and vagus nerve stimulation.

Considering strategies that are not recommended, the VA/DoD [32] does not recommend the use of vagal nerve stimulation. The authors point out that potential benefits could be overcome by common adverse effects of that procedure, including voice alteration, dysphagia, dyspnea, infection, dizziness, asthenia, chest pains, palpitations, or vocal cord paralysis. The CPG from NVL [33] has no specific non-recommendation except it does not recommend routine use of carbamazepine, lamotrigine, pindolol, valproate, dopamine agonists, psychostimulants or thyroid hormones for unipolar depression.

Regarding psychotherapy, only the CPG from NVL [33] highlights that psychotherapy should be one of the strategies for patients with TRD, while the VA/DoD [32] CPG says that this is a potentiation strategy for patients with partial or limited response to first-line treatment.

## Conclusions

We found that most recent high-quality CPGs with high-quality recommendations do not provide recommendations for the treatment of TRD. It is important to notice that there is a lack of consensus regarding the definition and clinical application of TRD. Here we adopted the definition supported by the STAR*D Trial, which considers TRD as a therapeutic failure after two adequate antidepressant therapies [6]. Moreover, we observed that among the two CPGs that provide approaches for the treatment of TRD, there are a lot of divergences regarding which strategy should be followed.

Finally, we have observed that for two CPGs that provide approaches for the treatment of TRD, there are divergences regarding which strategy should be followed. This situation may reflect differences in the definitions used, different approaches or even the moment in which the research was conducted. We understand that the scientific community should discuss this topic in order to better identify patients with resistant depression. In closing, there are still important misdiagnoses which are arguably leading to inadequate management of resistant depression.

## Supporting information

**S1 File. Search strategy.** The search strategy used in the PubMed (Medline), Embase, Cochrane Library, PsycINFO, and BVS databases.
(DOCX)

**S1 Fig. Flowchart of CPG selection.**
(DOCX)

## Acknowledgments

We thank Caroline Molino, Luciana Vasconcelos, and Nathália Celini Leite Santos for their invaluable assistance in helping us better understand the method of appraisal of CPGs.

## Author Contributions

**Conceptualization:** Franciele Cordeiro Gabriel, Airton Tetelbom Stein, Daniela de Oliveira Melo, Diogo Telles Correia, Eliane Ribeiro.

**Data curation:** Franciele Cordeiro Gabriel.

**Formal analysis:** Gessica Caroline Henrique Fontes-Mota, Itamires Benício dos Santos, Camila da Silva Rodrigues, Andrea Dourado, Renério Fráguas.

**Funding acquisition:** Eliane Ribeiro.

**Methodology:** Franciele Cordeiro Gabriel, Mônica Cristiane Rodrigues.

**Project administration:** Franciele Cordeiro Gabriel.

**Writing – original draft:** Franciele Cordeiro Gabriel, Airton Tetelbom Stein.

**Writing – review & editing:** Franciele Cordeiro Gabriel, Gessica Caroline Henrique Fontes-Mota, Itamires Benício dos Santos, Camila da Silva Rodrigues, Andrea Dourado, Renério Fráguas, Ivan D. Florez, Eliane Ribeiro.

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
