## [Decision Letter · Decision Letter 0]

9 Nov 2022

PONE-D-22-21981Guidelines’ recommendations for the treatment-resistant depression: a systematic review of their quality

PLOS ONE

Dear Dr. Gabriel,

Thank you for submitting your manuscript to PLOS ONE. After careful consideration, we feel that it has merit but does not fully meet PLOS ONE’s publication criteria as it currently stands with only minor revisions required. Therefore, we invite you to submit a revised version of the manuscript that addresses the points raised during the review process.

We look forward to receiving your revised manuscript.

Kind regards,

Michael Gilbert McCaul, MSc, PhD

Academic Editor

PLOS ONE

Journal Requirements:

Reviewers' comments:

Reviewer's Responses to Questions

**Comments to the Author**

1. Does the manuscript adhere to the experimental procedures and analyses described in the Registered Report Protocol?

If the manuscript reports any deviations from the planned experimental procedures and analyses, those must be reasonable and adequately justified.

Reviewer #1: Yes

Reviewer #2: Yes

2. If the manuscript reports exploratory analyses or experimental procedures not outlined in the original Registered Report Protocol, are these reasonable, justified and methodologically sound?

A Registered Report may include valid exploratory analyses not previously outlined in the Registered Report Protocol, as long as they are described as such.

Reviewer #1: Yes

Reviewer #2: Yes

3. Are the conclusions supported by the data and do they address the research question presented in the Registered Report Protocol?

The manuscript must describe a technically sound piece of scientific research with data that supports the conclusions. The conclusions must be drawn appropriately based on the research question(s) outlined in the Registered Report Protocol and on the data presented.

Reviewer #1: Yes

Reviewer #2: Yes

4. Have the authors made all data underlying the findings in their manuscript fully available?

Reviewer #1: Yes

Reviewer #2: No

5. Is the manuscript presented in an intelligible fashion and written in standard English?

Reviewer #1: Yes

Reviewer #2: No

6. Review Comments to the Author

Please use the space provided to explain your answers to the questions above. (Please upload your review as an attachment if it exceeds 20,000 characters)

Reviewer #1: Overall Comments

• This is a clear and well-written systematic review of the quality of guidelines’ recommendations for treatment-resistant depression. The authors provide an adequate rationale for conducting such an important study. They give sufficient explanations of the methods and analysis and key results that flowed well. I provide a few minor comments for possible edits in different sections of the manuscript.

Abstract

• Then/so – use one word instead. Line 57

• Consider summarising the results in the abstract – the details of which CPGs used specific terminologies can be described in the results section in the manuscript.

Introduction

• Consider using “people with depression” instead of “depressed people” – line 81

• Some of the studies you cited implicitly evaluated the quality of recommendations in the CPGs they reviewed. Consider rephrasing this line: “currently no study that has evaluated the quality of their recommendations”. Line 130-133.

Materials and methods

• Describe your exclusion criteria here. I see you mention some of these in the results section in Line 186-187.

Discussion

• You summarise the results again in the discussion section of the paper. The focus in this section should be a discussion of the implications of these differences/similarities or what they mean for clinical practice.

Reviewer #2: Thank you - this is a very interesting systematic review.

Some comments:

Line 147 to 150: please include the timeframe of the search.

Line 151: you should mention about excluding duplicates

Line 163 to 165: explain what each domain (i.e. domain 3 for AGREE II and domain 1 for AGREE-REX) are - you explained it

in the protocol but the reader should not need to go look up the protocol or the AGREE tools to understand which domain is being referred to.

Line 186: "7" should be "seven"

Table 1: explain abbreviations ("R", "NM", "CI") and the superscripts "#" and "*"

Line 242, 307, 466, 484: text in italics - should be normal text

Line 313: please reword this sentence - it does not read well. Consider something like "does not recommend routine use of carbamazepine..."

Line 331-2: Please clarify this sentence ("This situation...been conducted") - it is confusing

Line 333-7: Please reword this last sentence. It is long, repetitive and unclear

There are a number of grammatical errors (mostly punctuation) that should be corrected

7. PLOS authors have the option to publish the peer review history of their article (what does this mean?). If published, this will include your full peer review and any attached files.

Reviewer #1: No

Reviewer #2: No

---

## [Author Response · Author response to Decision Letter 0]

19 Dec 2022

November 23, 2022

Michael Gilbert McCaul, MSc, PhD

Academic Editor

PlosOne

Re: MS PONE-D-22-21981

Dear Dr McCaul,

Thank you very much for your e-mail of November 9, 2022, regarding our manuscript, PONE-D-22-21981. The reviewers’ comments were very helpful and significantly contributed to improve que quality of the manuscript’s revised version.

As requested, please find below the revised version of our paper, and the point-by-point list of revisions made. For your convenience, sections that were modified during the revision are highlighted in yellow in the revised version of the manuscript.

Reviewer #1: Overall Comments

• This is a clear and well-written systematic review of the quality of guidelines’ recommendations for treatment-resistant depression. The authors provide an adequate rationale for conducting such an important study. They give sufficient explanations of the methods and analysis and key results that flowed well. I provide a few minor comments for possible edits in different sections of the manuscript.

Reply: Thank you very much for your comments.

Abstract

• Then/so – use one word instead. Line 57

• Consider summarising the results in the abstract – the details of which CPGs used specific terminologies can be described in the results section in the manuscript.

Reply: A new summary was provided, which has the “then/so” expression corrected and included much more elements relating to the study results.

Introduction

• Consider using “people with depression” instead of “depressed people” – line 81

Reply: It has been corrected (line 77).

• Some of the studies you cited implicitly evaluated the quality of recommendations in the CPGs they reviewed. Consider rephrasing this line: “currently no study that has evaluated the quality of their recommendations”. Line 130-133.

Reply: Thanks for your comment. We rephrased this sentence to make it clear that we were referring to the fact that there were no papers evaluating CPGs’ recommendations specifically using the AGREE-REX instrument. (Lines 128-130)

Materials and methods

• Describe your exclusion criteria here. I see you mention some of these in the results section in Line 186-187.

Reply: Thanks for your comment. In addition of the mentioned study protocol, we introduced a short sentence in the methods section to briefly explain the study’s eligibility criteria. Please see lines 147-148.

Discussion

• You summarise the results again in the discussion section of the paper. The focus in this section should be a discussion of the implications of these differences/similarities or what they mean for clinical practice.

Reply: Thanks for your comment. We totally agree with your point. However, we have repeated some parts of the results in the discussion/conclusion sections in order to stress the urgency of the situation. We understand that a combined effort involving several stakeholders is required. As long as we have a lack of consensus for the definition itself (lines 276 – 279) and for the management strategies: 282-322), we’ll still continue to be unable to tackle this condition (lines: 337-339).

Reviewer #2: Thank you - this is a very interesting systematic review.

Reply: Thank you very much for your comments.

Some comments:

Line 147 to 150: please include the timeframe of the search.

Reply: Thanks for your comment. We have introduced this information. Please see lines 142-144.

Line 151: you should mention about excluding duplicates

Reply: Thanks for your comment. We have added this information on line 149.

Line 163 to 165: explain what each domain (i.e. domain 3 for AGREE II and domain 1 for AGREE-REX) are - you explained it in the protocol but the reader should not need to go look up the protocol or the AGREE tools to understand which domain is being referred to.

Reply: Thanks for your comment. We have included the denomination of each domain after it was mentioned. Please see lines 166-168.

Line 186: "7" should be "seven"

Reply: corrected.

Table 1: explain abbreviations ("R", "NM", "CI") and the superscripts "#" and "*"

Reply: the explanations have been included as table footnote.

Line 242, 307, 466, 484: text in italics - should be normal text.

Reply: Text in italics were converted to normal text.

Line 313: please reword this sentence - it does not read well. Consider something like "does not recommend routine use of carbamazepine..."

Reply: corrected.

Line 331-2: Please clarify this sentence ("This situation...been conducted") - it is confusing

Line 333-7: Please reword this last sentence. It is long, repetitive and unclear

Reply: The sentence has been changed. Please see lines 334-339.

There are a number of grammatical errors (mostly punctuation) that should be corrected

Reply: We sent the text to an English-speaking colleague to language revision.

We have addressed the comments and concerns listed by the Editor and the two expert Reviewers. The authors confirm that this manuscript has not been submitted to, and is not currently under review by, another journal. 

We therefore shall be grateful if you would consider our revised paper for publication in your Journal.

Thank you for your consideration. We look forward to hearing from you.

Sincerely,

Franciele Cordeiro Gabriel

Department of Pharmacy, Faculty of Pharmaceutical Sciences,

University of São Paulo, São Paulo Brazil

Email: francordegabriel@gmail.com

---

## [Decision Letter · Decision Letter 1]

25 Jan 2023

Guidelines’ recommendations for the treatment-resistant depression: a systematic review of their quality

PONE-D-22-21981R1

Dear Dr. Gabriel,

We’re pleased to inform you that your manuscript has been judged scientifically suitable for publication and will be formally accepted for publication once it meets all outstanding technical requirements.

Kind regards,

Norio Yasui-Furukori

Academic Editor

PLOS ONE

Additional Editor Comments (optional):

Reviewers' comments:

Reviewer's Responses to Questions

**Comments to the Author**

1. If the authors have adequately addressed your comments raised in a previous round of review and you feel that this manuscript is now acceptable for publication, you may indicate that here to bypass the “Comments to the Author” section, enter your conflict of interest statement in the “Confidential to Editor” section, and submit your "Accept" recommendation.

Reviewer #1: All comments have been addressed

Reviewer #2: All comments have been addressed

2. Is the manuscript technically sound, and do the data support the conclusions?

Reviewer #1: Yes

Reviewer #2: Yes

3. Has the statistical analysis been performed appropriately and rigorously? 

Reviewer #1: Yes

Reviewer #2: Yes

4. Have the authors made all data underlying the findings in their manuscript fully available?

Reviewer #1: Yes

Reviewer #2: Yes

5. Is the manuscript presented in an intelligible fashion and written in standard English?

Reviewer #1: Yes

Reviewer #2: Yes

6. Review Comments to the Author

Reviewer #1: The authors have addressed all my comments. I am satisfied with their edits, and I recommend that this article be accepted for publication.

Reviewer #2: Changes notes, review reads well and is very interesting. All my concerns have been addressed. Thanks.

7. PLOS authors have the option to publish the peer review history of their article (what does this mean?). If published, this will include your full peer review and any attached files.

Reviewer #1: No

Reviewer #2: No

---

## [Editor Report · Acceptance letter]

27 Jan 2023

PONE-D-22-21981R1 

Guidelines’ recommendations for the treatment-resistant depression: a systematic review of their quality 

Dear Dr. Gabriel:

I'm pleased to inform you that your manuscript has been deemed suitable for publication in PLOS ONE. Congratulations! Your manuscript is now with our production department. 

Kind regards, 

on behalf of

Dr. Norio Yasui-Furukori 

Academic Editor

PLOS ONE